# Cost-Effective Synthetic Data Generation for Post-Training using QWICK

## Abstract

Large language models (LLMs) are showing expert-level ability in various fields (e.g., programming and math). However, this progress heavily relies on the generation of high-quality synthetic data to improve the models' capabilities during post-training. Generating such data in a cost-effective manner presents a significant challenge. Specifically, stronger models tend to generate higher-quality data but come with a substantial computational cost, while weaker models are cheaper to run but may produce weaker outputs. In this paper, we introduce Question-Wise model pICK (QWICK) to address this challenge. By tracking the empirical reward, cost, and number of trials for each model, QWICK strikes a balance between exploitation and exploration, ultimately converging on a cost-effective model for each specific question. Specifically, QWICK achieves a 50% cost reduction on a programming dataset and a 40% cost reduction on a mathematics dataset, without compromising data quality. Furthermore, compared to baseline methods, our approach can produce up to 2.1 times more valid synthetic data at the same cost. Our anonymized code is available at https://anonymous.4open.science/r/QWICK-17C3

## 1 Introduction

In recent years, large language models (LLMs) have demonstrated notable success across various domains, even achieving silver-medal-level performance in the International Mathematics Olympiad (teams et al., 2024). The key to this success is post-training on domain-specific tasks and datasets, such as mathematics (Luo et al., 2023a; Tong et al., 2024; Xin et al., 2024a;b) and programming (Luo et al., 2023b). Traditionally, creating necessary post-training datasets relied on human annotations, a process that is both costly and time-consuming. To mitigate these challenges, Synthetic Data Generation (SDG) using state-of-the-art LLMs has emerged as a more scalable alternative – *autonomously* producing *large amounts* of high-quality data that reaches the level of human-generated ones (Gilardi et al., 2023; Singh et al., 2023; Bansal et al., 2024).

Despite these advantages, SDG faces challenges in balancing data quality with computational costs. Generating high-quality data typically demands substantial computational resources (Tong et al., 2024) or the use of high-performance, expensive LLMs. Conversely, using lower-cost models may generate lower-quality data, which risks degrading model performance or even causes catastrophic failure (Shumailov et al., 2024). For instance, OpenAI's o1 (OpenAI, 2024) charges $15 per million input tokens and $60 per million output tokens, whereas the Llama 70B model only costs between $0.35 to $1.00 per million tokens on various endpoints (together.ai, 2024; Deepinfra, 2024). Although using Llama 70B can cut costs by up to $\sim 150\times$ compared to OpenAI's o1, this cost reduction comes at the expense of data quality. This dilemma presents a critical research question:

*How can we cost-effectively generate high-quality synthetic data?*

To elucidate this problem, consider a typical data synthesis pipeline (Bansal et al., 2024; Tong et al., 2024), illustrated in Fig. 1, which begins with a seed dataset (e.g., MATH (Hendrycks et al., 2021)) containing question-answer pairs. The goal is to leverage many LLMs with varying inference costs and response quality to generate reasoning paths (i.e., model responses) for each question, thereby yielding a significantly expanded dataset of question-response pairs, which can be then used to train

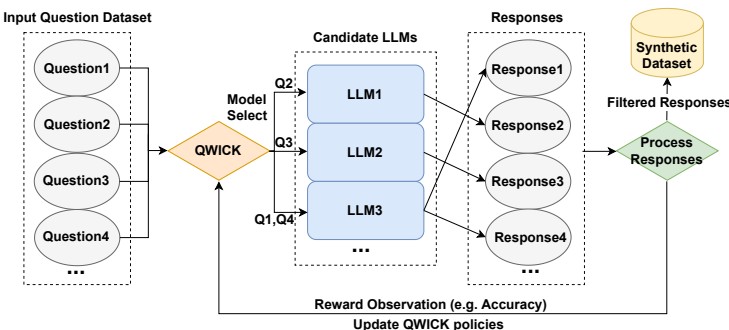

Figure 1: The SDG pipeline with QWICK for model selection. QWICK dynamically selects models for each question in the SDG process by balancing empirical *utility* and exploration. In each iteration, the algorithm processes the entire dataset, selects models to generate responses, observes the resulting rewards, and updates its internal statistics. This iterative process continues until the allocated budget is exhausted or a predefined stopping criterion is met.

and improve the model. A key part of this process is applying a threshold to filter out low-quality responses. For example, ground truth answers can be used to filter out synthetic responses that are incorrect. This makes model selection challenging. A stronger model may consistently pass the filtering step due to generating higher-quality responses but will also incur higher computational costs. In contrast, a weaker model might be cheaper to use but may produce a large number of un-qualified responses, ultimately wasting computational resources. A critical decision in this pipeline is, therefore, choosing the most appropriate model at each step

We propose to chose models based on an "utility" metric, defined as $utility = reward/cost$. Here, "cost" refers to the computational expenses per model call, and "reward" quantifies the model's contribution to the final synthetic dataset per model call. In the above case, we can apply a binary reward system: a reward of 1 is given when the model's generated response matches the ground truth, and 0 otherwise. Thus, the reward reflects the number of correct (valid) samples in the synthetic dataset. This reward system quantifies the model's contribution to the final synthetic dataset per model call. Note this is just an example and can be extended depending on the user's setting (e.g., using an outcome reward model), further detailed in §3.3. The model with the highest utility metric, balancing reward and cost, is considered the most cost-effective for the SDG pipeline.

However, identifying the most cost-effective model is challenging because the reward can only be determined through the SDG process itself, specific to each model and dataset. For example, in the binary reward setting, it is impossible to predict the average reward (i.e., accuracy) each model will achieve before the SDG process begins. Furthermore, estimating the *utility* (reward-to-cost ratio) for a given set of models on a particular dataset is even more challenging. Even if we have an accurate initial estimation of the utility of a model on a dataset, the utility can vary significantly across models for different portions of the dataset. That is, different models may perform best on different questions within the same dataset (§ 3.1). Therefore, selecting cost-effective models by collecting information *during* the SDG process becomes crucial.

To address this challenge, we propose Question-Wise model pICK (QWICK), which dynamically selects cost-effective models to generate synthetic data tailored to specific questions. QWICK uses a budget-limited question-wise multi-armed bandit (MAB) framework to balance exploiting well-performing models and exploring less-utilized ones. To illustrate, in QWICK, we are first given a list of models with varying costs and capability. During the SDG process, QWICK look at the cost and observed model reward, and employs a modified fractional KUBE algorithm to optimize model selection on the fly. This dynamic adjustment ensures highly cost-effective model selection throughout the SDG process.

We evaluate QWICK and show our method consistently outperforming baseline approaches in generated data quality, while spending lower cost throughout the SDG process. Specifically, our evaluation spans various model series (e.g., Gemma (Team et al., 2024), Llama (Dubey et al., 2024), Deepseek-Coder (Guo et al., 2024)) and domains (e.g., GSM8K (Cobbe et al., 2021),

MATH (Hendrycks et al., 2021), MBPP (Austin et al., 2021)), using different reward function setups such as binary and outcome-based reward models (Feng et al., 2023). Even without prior knowledge of the models' reasoning capabilities, QWICK consistently outperforms baseline approaches, delivering comparable data quality at up to 50% lower cost.

Our main contributions are summarized as follows:

- We introduce a budget-limited MAB algorithm QWICK for cost-effective SDG, utilizing a dynamic question-wise model selection strategy that adapts to ongoing assessments of model utility.
- We empirically validate that the proposed method outperforms the baselines not only in terms of reward metrics but also by producing a dataset that, when used for post-training, results in a model with higher accuracy at the same cost.

## 2 PROBLEM FORMULATION AND BACKGROUND

### 2.1 PROBLEM FORMULATION

To address the challenge of identifying cost-effective models for synthetic data generation on a dataset of input questions, we formulate the problem under budget constraints as the dynamic selection of the most cost-effective model, with the objective of maximizing the total reward. Please refer to Tab. 4 for all the notations below.

Given a question dataset $\mathcal{D} = \{\mathbf{x}_1, \ldots, \mathbf{x}_N\}$ containing $N$ input questions and a model pool $\mathcal{F} = \{f_1, \ldots, f_K\}$ consisting of $K$ language models, we define a policy $\pi(\mathbf{x})$ that selects a model to generate responses based on input question $\mathbf{x}$. For example, if $\pi(\mathbf{x}_1) = f_1$, model $f_1$ is selected to generate a response for the question $\mathbf{x}_1$. We adopt a multi-iteration model selection process. At each iteration $t$, the entire dataset $\mathcal{D}$ is processed, and a model is selected for each $\mathbf{x}_j$, where $1 \leq j \leq N$. The model selection policy $\pi_t(\mathbf{x}_j)$ is updated at each iteration $t$ for each question. After each model selection, we obtain a response $\mathbf{o}_{j,t}$ for the corresponding question. Once a response $\mathbf{o}_{j,t}$ is generated at iteration $t$, a cost $c_{\pi_t(\mathbf{x}_j),t,j}$ is incurred, and a reward $r_{\pi_t(\mathbf{x}_j),t} \in [0, 1]$ is observed, which represents the quality of the response generated by the selected language model for question $\mathbf{x}_j$. Let $\mathcal{G}(\pi)$ represent the expected total reward obtained by policy $\pi$. Our objective is to approximate the optimal policy $\pi^*$ that maximizes the $\mathcal{G}(\pi)$ while adhering to a budget constraint $B$:

$$\pi^* = \arg\max_\pi \mathcal{G}(\pi) = \arg\max_\pi \sum_t \sum_{j=1}^N r_{\pi_t(\mathbf{x}_j),t} : \mathcal{B} \geq \sum_t \sum_{j=1}^N c_{\pi_t(\mathbf{x}_j),t,j} \tag{1}$$

### 2.2 BUDGET-LIMITED MULTI-ARMED BANDITS

The multi-armed bandit (MAB) problem is a classic framework used to balance exploration and exploitation in decision-making (Robbins, 1952). Several strategies exist to address this problem, including $\epsilon$-Greedy (Sutton & Barto, 2018), Thompson Sampling (Chapelle & Li, 2011), Upper Confidence Bound (see Algorithm 3). An important extension of the MAB problem is the budget-limited MAB, also known as Bandits with Knapsacks (Tran-Thanh et al., 2010). One notable solution is the fractional KUBE algorithm (Tran-Thanh et al., 2012).

In the budget-limited MAB, there are $K$ arms and a total budget $\mathcal{B}$. At each iteration $t$, the algorithm pulls the arm $i$ selected by the policy $\pi_t$, then the cost $c_{i,t}$ and the reward $r_{i,t}$ are observed. The budget $\mathcal{B}$ is then reduced by $c_{i,t}$. The process continues until the budget is exhausted. The objective is to maximize the total reward obtained by the time the budget is exhausted.

The fractional KUBE (Algorithm 2) tracks the empirical mean reward $\hat{r}_{i,t}$, which is the average of the observed rewards $r_{i,t}$ for arm $i$ ($1 \leq i \leq K$), and the number of times arm $i$ has been pulled, denoted as $n_{i,t}$, up to iteration $t$. For $t < K$, each arm is pulled once in turn. For $t \geq K$, the arm with the highest utility, defined as $\pi_t = \arg\max_i \left( \frac{\hat{r}_{i,t}}{c_{i,t}} + \frac{1}{c_{i,t}} \sqrt{\frac{2 \ln t}{n_{i,t}}} \right)$, is selected.

The core insight of fractional KUBE is to exploit the arms with the highest empirical *utility* (the reward-cost ratio), instead of the highest reward. This enables fractional KUBE to find a policy

$\pi$ that minimizes total regret $\mathcal{R}(\pi) = \mathcal{G}(\pi^*) - \mathcal{G}(\pi)$ under budget $\mathcal{B}$. Here, $\mathcal{G}(\pi)$ represents the difference in rewards between the optimal policy $\pi^*$ and policy $\pi$. Note that $\pi^*$ always selects the arm with the highest *utility* (i.e., the reward-to-cost ratio). Next, we will leverage this algorithm to address our problem.

## 3 METHOD

We propose the Question-Wise model pICK (QWICK) Algorithm (detailed in Algorithm 1), with the full pipeline illustrated in Fig. 1. This algorithm is designed to optimize the reward (e.g. the amount of valid data) in synthetic data generation by dynamically selecting the best model for each question under the problem formulation (§ 2). The algorithm effectively finds the Pareto frontier (see Fig.2), ensuring the language model with the highest *utility* is selected for each question $\mathbf{x}_j \in \mathcal{D}$ while adhering the budget constraint $\mathcal{B}$ (§2.1).

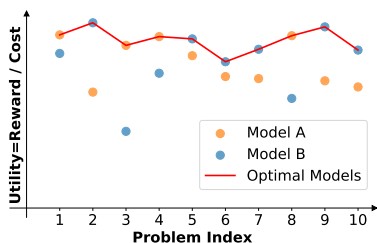

Figure 2: The fractional KUBE identifies the Pareto frontier.

### 3.1 QUESTION-WISE MODEL PICK

Inspired by fractional KUBE, we consider *utility* (reward-cost ratio) as a crucial factor in determining which model to use. We base our question-wise model pick on a simple yet important observation: models can exhibit varing performance across the entire dataset. Specifically, some models excel in terms of *utility* on certain questions, while others perform better on different ones, as illustrated in Tab. 1. This variability is observed both within models of different sizes from the same family (e.g., Gemma-2 (Team et al., 2024)) and across different model series (e.g., Gemma-2 (Team et al., 2024), Llama-3.1 (Dubey et al., 2024), and Phi-3 (Abdin et al., 2024)).

Table 1: Proportion of instances where different models perform the best in *utility* on a specific problem across the entire MATH (Hendrycks et al., 2021) dataset.

|  | Gemma | Phi | Llama |
|---|---|---|---|
| **Share (%)** | 9.13% | 35.43% | 55.44% |

|  | Gemma 2B | Gemma 9B | Gemma 27B |
|---|---|---|---|
| **Share (%)** | 41.16% | 47.00% | 11.84% |

Based on this observation, we opt to select models on a per-question basis rather than relying on a single model for all questions. In each iteration, we evaluate the dataset and assign the best model to each question based on its empirical utility and the number of trials, then generate synthetic responses accordingly. The following section describes the algorithm in detail.

### 3.2 UTILITY-DRIVEN QUESTION-WISE MODEL PICK FOR SYNTHETIC DATA GENERATION

The algorithm (Algorithm 1) takes the question dataset $\mathcal{D}$ of size $N$ and the model pool $\mathcal{F}$ of size $K$ as inputs. The models in the model pool are indexed in increasing order of per-token inference cost, denoted as $a_i$, with $1 \le i \le K$. For each input question $\mathbf{x}_j$ from the input dataset $\mathcal{D}$ with $1 \le j \le N$, the algorithm maintains a model pool $P(\mathbf{x}_j)$, initially containing only the cheapest model $f_1$. The following multi-iteration model selection and response generation process continues until the stopping condition for each question is met or the budget $\mathcal{B}$ is exhausted.

At each iteration, the algorithm processes all question inputs $\mathbf{x}_j$ in the question dataset $\mathcal{D}$. For any question, if its model pool (with size $l$) contains fewer than $K$ models (i.e., $l < K$), we compare the highest empirical reward-to-cost ratio (*utility*) in the current pool (i.e. $\max_{i \in P(\mathbf{x}_j)} \left( \frac{\hat{r}_{i,t,j}}{a_i} \right)$) with the maximum potential reward-to-cost ratio of the next model in line, assuming a reward of 1 for that model (i.e., $\frac{1}{a_{l+1}}$) (line 16). If the potential ratio of the next model is greater, we add it to the pool and select it for the next attempt. Otherwise, the algorithm defaults to selecting a model by balancing exploration and exploitation within the existing pool. This approach enables the algorithm to stop further exploration when the potential rewards of models outside the pool fall below the current maximum empirical rewards within the pool. This reduces excessive exploration

---

**Algorithm 1** Question-Wise model pICK (QWICK) Algorithm

---

1: **Input:** Budget $B$, question dataset $\mathcal{D}$, stopping condition $Stop(\mathbf{x}_j)$ for each question $\mathbf{x}_j \in \mathcal{D}$
2: **Input:** $K$ models, where the $i$-th model is $f_i$. Inference cost per token for model $f_i$ is $a_i$ $(1 \leq i \leq K)$. Models are sorted in increasing order of $a_i$.
3: **Input:** $\beta$ is a weight for balancing question-wise utility and dataset-level utility in model selection. $\alpha$ is a weight controlling the exploration term.
4: **Environment:** At iteration $t$, for a given question $\mathbf{x}_j$, model $f_i$ is selected by the action $\pi_t(\mathbf{x}_j)$ (denoted as $\pi_{t,j}$). The observed reward is $r_{i,t} \in [0,1]$, and the cost is $c_{i,t,j}$. The empirical mean reward of $f_i$ for $\mathbf{x}_j$ is $\hat{r}_{i,t,j}$. The empirical mean reward of $f_i$ over the entire dataset $\mathcal{D}$ is $\hat{r}_{i,t}$. The empirical normalized cost of querying $f_i$ for question $\mathbf{x}_j$ is $\hat{c}_{i,t,j}$. The number of trials using $f_i$ for $\mathbf{x}_j$ until iteration $t$ is $n_{i,t,j}$.
5: **Initialize:** $t \leftarrow 1$
6: **Initialize:** Remaining budget $\mathcal{B}_t \leftarrow \mathcal{B}$
7: **Initialize:** Model pool for each question $P(\mathbf{x}_j) \leftarrow [1]$ for all $\mathbf{x}_j \in \mathcal{D}$
8: **while** $\mathcal{D} \neq \emptyset$ **do**
9:   **for** $\mathbf{x}_j \in \mathcal{D}$ **do**
10:     **if** $Stop(\mathbf{x}_j)$ **then**
11:       Remove $\mathbf{x}_j$ from $\mathcal{D}$ {Remove question $\mathbf{x}_j$ that meets the stopping condition}
12:       **continue**
13:     **end if**
14:     $l \leftarrow \text{len}(P(\mathbf{x}_j))$
15:     **if** $l < K$ **then**
16:       **if** $\max_{i \in P(\mathbf{x}_j)} \left( \frac{\hat{r}_{i,t,j}}{a_i} \right) < \frac{1}{a_{l+1}}$ **then**
17:         $\pi_{t,j} \leftarrow l+1$ {Select the next higher-cost model}
18:         Append $l+1$ to $P(\mathbf{x}_j)$ {Add the selected model to the pool for question $\mathbf{x}_j$}
19:       **else**
20:         $\pi_{t,j} \leftarrow \operatorname{argmax}_{i \in P(\mathbf{x}_j)} \left( \frac{\min_{i' \in P(\mathbf{x}_j)} \hat{c}_{i',t,j}}{\hat{c}_{i,t,j}} \left( \beta \hat{r}_{i,t,j} + (1-\beta)\hat{r}_{i,t} \right) + \frac{1}{\alpha} \sqrt{\frac{2 \ln t}{n_{i,t,j}}} \right)$
        {Select the best model based on estimated rewards and exploration term}
21:       **end if**
22:     **else**
23:       $\pi_{t,j} \leftarrow \operatorname{argmax}_{i \in P(\mathbf{x}_j)} \left( \frac{\min_{i' \in P(\mathbf{x}_j)} \hat{c}_{i',t,j}}{\hat{c}_{i,t,j}} \left( \beta \hat{r}_{i,t,j} + (1-\beta)\hat{r}_{i,t} \right) + \frac{1}{\alpha} \sqrt{\frac{2 \ln t}{n_{i,t,j}}} \right)$ {Select the best model based on estimated rewards and exploration term}
24:     **end if**
25:     Update remaining budget $\mathcal{B}_t \leftarrow \mathcal{B}_t - c_{\pi_{t,j},t,j}$
26:     **if** $B_t < 0$ **then**
27:       **Exit** {Terminate if budget is exhausted}
28:     **end if**
29:     Use model $f_{\pi_{t,j}}$ to generate response and observe the reward $r_{\pi_{t,j},t}$
30:     Update the estimated reward $\hat{r}_{\pi_{t,j},t,j}$, and $\hat{r}_{\pi_{t,j},t}$, the cost $\hat{c}_{\pi_{t,j},t,j}$, and the number of pulls $n_{\pi_{t,j},t,j}$ {Update statistics for the selected model}
31:   **end for**
32:   $\mathcal{B}_{t+1} \leftarrow \mathcal{B}_t$
33:   $t \leftarrow t+1$
34: **end while**

---

commonly associated with traditional algorithms. Note that we assume uniform generation lengths across models, as only the per-token cost $a_i$ is used to estimate the reward-to-cost ratio.

If the model pool already contains all $K$ models, the algorithm selects the model $i$ that maximizes the expression $\frac{\min_{i' \in P(\mathbf{x}_j)} \hat{c}_{i',t,j}}{\hat{c}_{i,t,j}} \left( \beta \hat{r}_{i,t,j} + (1-\beta)\hat{r}_{i,t} \right) + \frac{1}{\alpha} \sqrt{\frac{2 \ln t}{n_{i,t,j}}}$ (line 23). The first term balances question-level utility with dataset-level utility by mixing the question-level reward $\hat{r}_{i,t,j}$ with the dataset-level reward $\hat{r}_{i,t}$. Without loss of generality, the scaling factor $\frac{\min_{i' \in P(\mathbf{x_i})} \hat{c}_{i',t,i}}{\hat{c}_{i,t,j}}$ normalizes the first term to the range [0,1]. The second term, $\frac{1}{\alpha} \sqrt{\frac{2 \ln t}{n_{i,t,j}}}$, encourages exploration of underused models. In our evaluations, we simply set $\alpha = 16$ and $\beta = 0.5$ to balance between question-level

utility, dataset-level utility, and the trade-off between exploration and exploitation. Note that that $\hat{c}$ is used to estimate the cost, as the true cost is unknown before each generation process.

The algorithm proceeds iteratively, and when the stopping condition for a specific question $\mathbf{x}_j$ ($1 \leq j \leq N$) is met (such as reaching a target number of correct answers or hitting the inference cost threshold), that question is removed from $\mathcal{D}$. The outer loop terminates when either no more questions remain in $\mathcal{D}$ or the budget is depleted.

### 3.3 FLEXIBLE UTILITY METRIC

The *utility* (reward-cost ratio) metric is flexible in the proposed algorithm to accommodate diverse use cases. This flexibility operates on two levels. First, the cost is easily adjustable by factoring in the per-token pricing provided by the LLM service provider. Second, the reward component is also configurable. For example, in tasks like math or code where the ground truth is available, we can verify if the generated answer matches the correct one. If the answer is correct, a reward of 1 is assigned; otherwise, the reward is 0. Furthermore, a more granular reward system can be implemented using an Outcome Reward Model (ORM), which assigns a score between 0 and 1, where higher values reflect better answer quality. Our evaluations (in §4.1 and §4.3) demonstrate that the proposed method achieves higher rewards within the same budget when using varied utility metrics.

### 4 EXPERIMENTS

**Methodology.** To assess the effectiveness of the proposed method across various scenarios, we conducted evaluations on both math (GSM8K (Cobbe et al., 2021) and MATH (Hendrycks et al., 2021)) and programming (MBPP (Austin et al., 2021)) tasks. These datasets include both questions and ground truth answers or test cases. We used the QWICK and baseline methods to generate synthetic responses for the questions in the evaluation dataset under different budget settings. We then evaluated the quality of the synthetic datasets in terms of diversity and coverage. Additionally, we fine-tuned a model using the synthetic datasets and tested the fine-tuned model on the corresponding test datasets. This approach provided a comprehensive assessment of the synthetic dataset quality under various methods within a constrained budget.

**Inference Settings.** *For synthetic data generation*, we generate responses by inputting questions from the MATH, GSM8K, and MBPP datasets into a list of corresponding LLMs. We use models with varying computational costs and capabilities, achieved by using different model sizes from the same series for each dataset, as detailed in Tab. 2. We set the temperature to 1 and limit the maximum token generation to 2048. To ensure the quality of generated responses, we apply reject sampling (Yuan et al., 2023) to filter out incorrect responses. For math tasks, the generated answers were compared against the ground truth, while for programming tasks, we executed the generated code and filtered out responses that failed to execute or did not pass the test cases. To achieve uniform sampling across the dataset, we set a maximum number of valid responses per question, as outlined in Tab.2, following the approach in Tong et al. (2024). *For evaluation*, we generate responses using the fine-tuned models with greedy sampling, setting a token limit of 2048. We evaluate model performance using pass@1, where only the first generated response is considered, and report accuracy for all experiments. Additionally, we apply Chain-of-Thought (CoT) prompting (Wei et al., 2022) to enhance reasoning in both synthetic data generation and evaluation.

**Fine-tuning Settings.** We fine-tuned each model on the generated datasets, running 200 steps for the GSM8K and MATH datasets, and 20 epochs for the MBPP dataset. For the math tasks, checkpoints were saved every 20 steps, while for the programming tasks, checkpoints were saved every 5 steps. We report the highest accuracy achieved across all checkpoints. Instruction tuning was employed for fine-tuning, with a batch size of 64, utilizing Sequence Packing (Krell et al., 2021) to reduce the total number of fine-tuning steps, following Tong et al. (2024). Fine-tuning was performed on 2 A100 80GB GPUs with a gradient accumulation size of 16. We used the Adam optimizer with no weight decay, combined with a cosine learning rate scheduler. For the programming tasks, we fine-tuned the Llama-2-7B (Touvron et al., 2023) model, and for the math tasks, we fine-tuned the Llama-3-8B (Meta, 2024) model, both with a maximum learning rate of 5e-5.

**Utility Calculation.** The cost of synthetic data generation is calculated on a per-token basis, primarily estimated according to model size, following the pricing structure of TogetherAI's serverless endpoints (together.ai, 2024). For each response, the cost is determined by multiplying the number of generated tokens by the per-token price. The reward calculation is binary: it is set to 1 if the generated answer matches the ground truth (i.e., it is a valid sample), and 0 otherwise. The total reward, therefore, reflects the number of valid samples.

**Baseline Settings.** We compared the proposed QWICK algorithm against following two baseline settings:

- **Random Model Selection (Algorithm 5).** When prior knowledge of a model's performance on a specific dataset is unavailable, a straightforward approach is to randomly select a model for each question. In this setting, we applied uniform random selection, where a model is chosen randomly for each question.

- **Dataset-wise UCB1 (Algorithm 4).** We adapted the classic UCB1 algorithm (Algorithm 3) to select the model based on upper confidence bound on the reward for the entire dataset at each iteration. The process continued until the budget, $\mathcal{B}$, was fully exhausted. Like the original UCB1, this adapted version focuses on maximizing the reward but does not take into account the cost associated with model calls. Instead, it prioritizes selecting the model that is expected to yield the highest cumulative reward, without considering the cost of achieving that reward.

Table 2: Dataset and model settings for §4.1

| Dataset | Model Type | Model List | Max #Response Per Question |
|---|---|---|---|
| GSM8K (Cobbe et al., 2021) | Llama-3.1 (Dubey et al., 2024) | 8B, 70B | 3 |
| MATH (Hendrycks et al., 2021) | Gemma-2 (Team et al., 2024) | 2B, 9B, 27B | 10 |
| MBPP (Austin et al., 2021) | Deepseek-Coder (Guo et al., 2024) | 1.3B, 6.7B, 33B | 10 |

## 4.1 MAIN RESULTS

We demonstrate that our method can generate synthetic datasets of comparable quality at a lower cost across various question datasets. Specifically, we fine-tune the Llama-3-8B model on the GSM8K and MATH datasets and the Llama-2-7B model on the MBPP dataset, using synthetic datasets generated by different methods. We then report the accuracy of these fine-tuned models on their respective test sets, as a measure of synthetic dataset quality. As shown in the first row of Fig. 4, QWICK achieves comparable or identical accuracy with up to 40% lower cost on GSM8K, up to 33% lower cost on MATH, and up to 50% lower cost on MBPP compared to the UCB1 method.

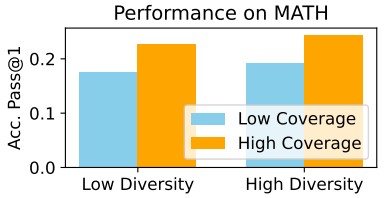

Figure 3: Coverage and diversity can positively boost the accuracy.

These performance gains are largely attributed to the increased diversity and coverage of the synthetic datasets. As illustrated in Fig. 3 and supported by Bansal et al. (2024), synthetic dataset with greater diversity and broader coverage enable models fine-tuned on these datasets to achieve higher test accuracy. Specifically, QWICK consistently outperforms the baselines on these datasets in both diversity and coverage metrics. For instance, as shown in the second row of Fig.4, QWICK generates up to 69%, 112%, and 106% more valid samples on GSM8K, MATH, and MBPP, respectively, compared to UCB1. Moreover, the third row of Fig.4 demonstrates that QWICK consistently maintains higher coverage than baseline methods across all these datasets under different cost constraints.

Note that the total reward is equivalent to the number of valid samples, as the reward is binary. The dataset-wise UCB1 algorithm focuses on maximizing the reward but neglects the associated costs, which hinders its ability to identify the most *cost-effective* model. In fact, in some cases, UCB1 performs worse than random selection due to this oversight. In contrast, QWICK successfully maximizes the reward within a given budget by identifying the cost-effective model for each question.

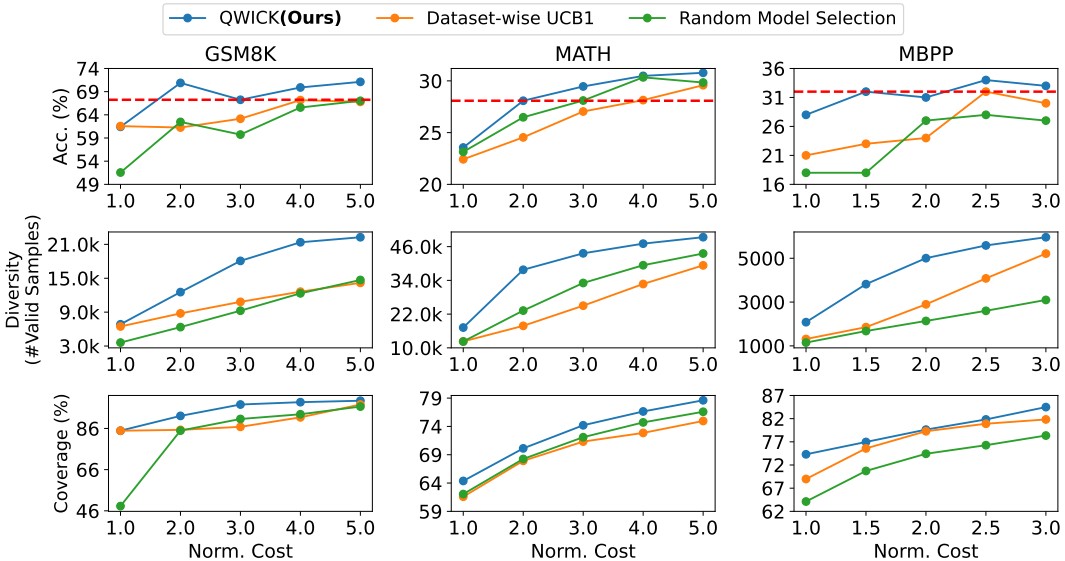

Figure 4: Accuracy, diversity, and coverage comparisons on GSM8K, MATH, and MBPP datasets with different costs with QWICK and baselines.

## 4.2 ANALYSIS OF EFFECTIVENESS

We demonstrate QWICK's model selection process on the MATH dataset using Gemma models (2b, 9b, and 27b) to illustrate its model selection convergence trace. The algorithm starts with the least expensive model (i.e., Gemma-2-2b) and progressively switches to larger models on questions where Gemma-2-2b performs poorly. After a few iterations, it converges on the most cost-effective model for most questions with potential solutions, as depicted in Fig. 5a. In contrast, a dataset-level model selection algorithm will converge to a single model for the entire dataset after a few iterations (e.g., 4 iterations), depending on the policy applied. For instance, an accuracy-driven algorithm (e.g., dataset-wise UCB1) will repeatedly select the Gemma-2-27B model, while an utility-driven algorithm will favor the Gemma-2-2B model. However, these models are sub-optimal when evaluated on a per-question basis, resulting in lower overall reward (measured by the number of valid samples in this case) and poorer coverage. We illustrate the total reward and coverage for these settings with the same maximum answer limit per question and the same cost limit as in Fig. 5b. The proposed method outperforms the baselines on both metrics.

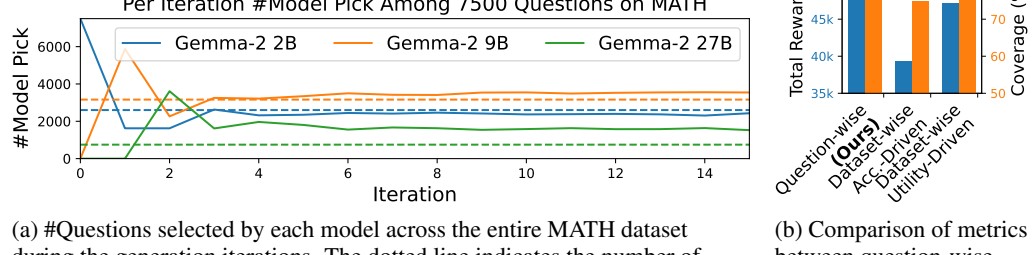

(a) #Questions selected by each model across the entire MATH dataset during the generation iterations. The dotted line indicates the number of questions for which a model is utility-optimal ($\phi^*$), after excluding those for which no correct solutions were generated. We set no maximum number of responses per question and set $\beta = 1$ to allow for clearer illustration.

(b) Comparison of metrics between question-wise and dataset-wise methods for the results on MATH in §4.1.

Figure 5: Visualizing the effectiveness of QWICK

## 4.3 ABLATION STUDY

**Generalization of the *utility* metric.** We demonstrate that the *utility* metric can be applied to a broader range of use cases. In §4.1, the reward is binary. However, this approach overlooks incorrect reasoning paths and does not account for varying answer quality. To address this, we utilize an Outcome Reward Model (ORM) fine-tuned on GSM8K by Feng et al. (2023), which allows for more nuanced reward assignment. The ORM assigns a score between $[-1, 1]$, which we linearly map to the range of $[0, 1]$ to align with the reward scale in the algorithm. Besides, we enforce a reward of 0 if the answer does not match the ground truth. As shown in Fig. 6, our method outperforms UCB1 and random selection across accuracy, diversity, and coverage metrics. In terms of total reward, QWICK achieves up to 2.2x higher results compared to UCB1 under the same budgets.

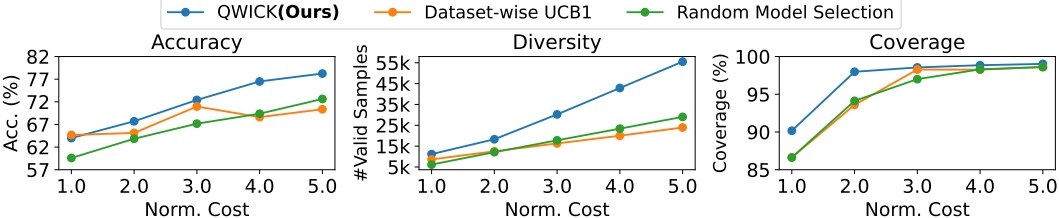

Figure 6: Accuracy, diversity, and coverage comparison on GSM8K with rewards by an ORM. The maximum number of responses per question is set to 10.

**Generalization of the synthetic dataset across different models.** To evaluate the generalization across different models being fine-tuned of the synthetic dataset quality produced by the proposed methods, we fine-tuned the v0.3 version of Mistral 7B model (Jiang et al., 2023), adjusting the maximum learning rate to 1e-5, using datasets generated by different methods and measured the model's accuracy on the test set. The results, shown in Tab. 3, demonstrate that the proposed method, QWICK, achieves similar accuracy with up to 66.6% lower cost compared with both UCB1 and random model selection when fine-tuning the Mistral 7B model, indicating that it is effective beyond just the Llama model.

Table 3: Accuracy comparisons on GSM8K fine-tuning Mistral 7B with different synthetic dataset

| Norm. Cost | 1X | 2X | 3X | 4X | 5X |
|---|---|---|---|---|---|
| **QWICK** | **67.1%** | **68.2%** | **69.5%** | **71.8%** | **73.4%** |
| UCB1 | 64.0% | 66.7% | 65.6% | 71.2% | 68.7% |
| Random | 59.4% | 66.1% | 67.0% | 68.9% | 69.7% |

**Generalization of the synthetic dataset across different reasoning method.** We utilize the Tool-Integrated Reasoning Agent (ToRA) by Gou et al. (2024) instead of the simpler CoT approach to generate synthetic data. This is done to evaluate the generalization capabilities of the method across different reasoning frameworks. Synthetic datasets were created using the 3B, 7B, and 14B Qwen2.5 models (Team, 2024) on the MATH dataset, employing ToRA along with various model selection strategies: random, dataset-wise UCB1, and QWICK. Each correct response generated by these models during the data creation phase was awarded a reward of 1, with incorrect responses receiving a reward of 0. Subsequently, these datasets were used to fine-tune a Llama-3-8B model over three epochs. The diversity and coverage of the synthetic dataset and the accuracy of the fine-tuned model on the test set are illustrated in Fig. 7. QWICK demonstrated a potential to reduce costs by up to 60% while achieving comparable accuracy to that obtained using the UCB1 and random model selection methods.

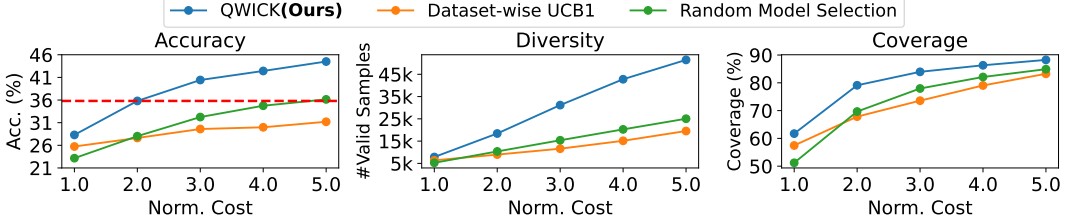

Figure 7: Accuracy, diversity, and coverage comparison on MATH with ToRA. The maximum number of responses per question is set to 10

## 5 RELATED WORK

**Cost-Effective Sampling.** Recent research has focused on combining search algorithms (e.g., Xin et al. (2024b), Xie et al. (2024) Yao et al. (2024)) with small yet strong language models ( OpenAI (2024); Xin et al. (2024b)) to achieve cost-effective performance. Other works study the trade-off between compute budget, model scale, and problem-solving performance at test time (Snell et al. (2024), Wu et al. (2024)). Furthermore, research has shown the effectiveness of synthetic data generated by small language models for fine-tuning stronger reasoners in supervised tasks, such as math and coding ( Bansal et al. (2024)). While smaller models typically perform better under fixed costs, larger models offer superior data quality, and performance can vary even among models of the same size. Selecting the most cost-effective model or combination for a given dataset remains challenging. Our work builds on previous approaches by proposing an algorithm that inherently achieves cost-efficient sampling.

**Learning LLM Reasoning.** Several studies have investigated how to enhance the reasoning capabilities of large language models (LLMs) using synthetic data in fine-tuning ( Yuan et al. (2023); Gulcehre et al. (2023); Wu et al. (2024)). A common strategy is to aggregate diverse reasoning paths generated through repeated sampling ( Wang et al. (2022); Li et al. (2023)). Some studies have utilized rejection sampling in combination with repeated sampling to filter diverse reasoning paths for math dataset augmentation in the post-training phase (Zelikman et al. (2022); Yuan et al. (2023); Tong et al. (2024)). Researchers have also explored reinforcement learning techniques to further improve the mathematical reasoning skills of LLMs, drawing distinctions between outcome-based and process-based reward models (Uesato et al. (2022); Lightman et al. (2024); Chen et al. (2024)). In our work, we focus on a streamlined method for generating augmented samples via outcome rejection sampling.

**Online Model Selection.** Online model selection is important for selecting the best-performing models from a set, especially given limited training resources and performance evaluations. Research in LLM model selection predominantly focuses on two areas: (1) selecting the best performing model during inference (Ong et al. (2024); Peng et al. (2023)) and (2) non-stationary selection, which accounts for changes in model performance due to iterative fine-tuning ( Xia et al. (2024)). However, these studies have not explored how to optimize model selection under budget constraints, which is formulated as knapsack-based multi-armed bandit problem. Methods such as fractional KUBE ( Tran-Thanh et al. (2012)) and budgeted Thompson sampling ( Xia et al. (2015)) have been developed for this task. The challenge extends to synthetic data generation as well. For instance, Luo et al. (2024) proposed an approach where all models are evaluated to determine the best model-answer pairs. This process can be streamlined using online model selection, by narrowing down the top-performing models at each inference step.

## 6 DISCUSSION

**Limitation.** QWICK maximizes the total reward within a budget constraint. However, determining how to accurately measure this total reward is non-trivial. Both the binary reward (0 or 1) and rewards based on an Outcome Reward Model (ORM) have limitations. The former ignores important factors such as coherence, completeness, and conciseness in reasoning, while the latter heavily relies on the quality of the ORM itself.

**Future work.** Future work could explore the use of Process Reward Models and more advanced search algorithms to generate higher-quality reasoning data. Additionally, experimenting with more effective post-training techniques may further improve outcomes.

## 7 CONCLUSION

In this paper, we propose QWICK, an cost-effective synthetic data generation framework for post-training through question-wise model selection. QWICK employs utility-driven model selection by framing the problem as a multi-armed bandit with budget constraints. Our evaluations on math and programming tasks demonstrate that this method can reduce costs by up to 50% while maintaining comparable dataset quality to baseline approaches.

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

# A    NOTATIONS

| Notation | Definition |
|---|---|
| $\mathcal{D} = \{\mathbf{x}_1, \ldots, \mathbf{x}_N\}$ | dataset |
| $\mathcal{F} = \{f_1, \ldots, f_K\}$ | collection of arms |
| $\mathcal{B}$ | total budget |
| $N$ | dataset size |
| $K$ | number of arms |
| $\pi_t(\mathbf{x})$ | model selection policy on input $\mathbf{x}$ at iteration $t$ |
| $r_{i,t}$ | reward of pulling arm $i$ at iteration $t$ |
| $c_{i,t}$ | cost of pulling arm $i$ at iteration $t$ |
| $a_i$ | per token cost of the LLM $i$ |
| $\mathcal{G}(\pi)$ | expected total reward earned by using $\pi$ to pull the arms |
| $\mathcal{R}(\pi)$ | regret of $\pi$ |

Table 4: Explainations of the notations

# B    ALGORITHMS

## B.1    FRACTIONAL KUBE (FOR KNAPSACK–BASED UPPER CONFIDENCE BOUND EXPLORATION AND EXPLOITATION)

---

**Algorithm 2** Fractional KUBE by Tran-Thanh et al. (2012)

---

1: **Input:** Budget $\mathcal{B}$, number of arms $K$
2: **Environment:** At each iteration $t$, we pull an arm $\pi_t$. The cost of pulling arm $i$ at time $t$ is $c_{i,t}$, and the reward received is $r_{i,t}$. The empirical mean reward of arm $i$ up to time $t$ is $\hat{r}_{i,t}$, and the total number of pulls for arm $i$ up to time $t$ is $n_{i,t}$. This holds for $1 \leq i \leq K$.
3: **Initialize:** $t \leftarrow 1$
4: **Initialize:** remaining budget $\mathcal{B}_t \leftarrow \mathcal{B}$
5: **while** True **do**
6:    **if** $\mathcal{B}_t < \min_{1 \leq i \leq K} c_{i,t}$ **then**
7:       **break** {Stop if the remaining budget is less than the minimum arm cost}
8:    **end if**
9:    **if** $t \leq k$ **then**
10:       $\pi_t \leftarrow t$ {Pull each arm once during the first $K$ iterations}
11:    **else**
12:       $\pi_t \leftarrow \operatorname{argmax}_i \left( \frac{\hat{r}_{i,t}}{c_{i,t}} + \frac{1}{c_{i,t}} \sqrt{\frac{2 \ln t}{n_{i,t}}} \right)$ {Select the arm that maximizes the estimated reward-to-cost ratio with exploration adjustment}
13:    **end if**
14:    Pull arm $\pi_t$ and observe the reward $r_{\pi_t, t}$
15:    Update the estimated reward $\hat{r}_{\pi_t, t}$ and the number of pulls $n_{\pi_t, t}$
16:    $\mathcal{B}_{t+1} \leftarrow \mathcal{B}_t - c_{\pi_t, t}$ {Deduct the cost of the selected arm from the remaining budget}
17:    $t \leftarrow t + 1$
18: **end while**

---

## B.2 UCB1 (Upper Confidence Bound version1)

---

**Algorithm 3** UCB1 by AUER et al. (2002)

---

1: **Input:** number of arms $K$
2: **Environment:** At each iteration $t$, an arm $\pi_t$ is pulled. The reward received from pulling arm $i$ at iteration $t$ is $r_{i,t}$. The empirical mean reward for arm $i$ up to iteration $t$ is $\hat{r}_{i,t}$. The total number of times arm $i$ has been pulled until iteration $t$ is $n_{i,t}$. This holds for $1 \leq i \leq K$.
3: **Initialize:** $t \leftarrow 1$
4: **while** True **do**
5:    **if** $t \leq k$ **then**
6:       $\pi_t \leftarrow t$ {Pull each arm once in the first $K$ iterations}
7:    **else**
8:       $\pi_t \leftarrow \text{argmax}_i \left( \hat{r}_{i,t} + \sqrt{\frac{2 \ln t}{n_{i,t}}} \right)$ {Select the arm that maximizes the upper confidence bound}
9:    **end if**
10:   Pull arm $\pi_t$ and observe the reward $r_{\pi_t,t}$
11:   Update the empirical mean reward $\hat{r}_{\pi_t,t}$ and the number of pulls $n_{\pi_t,t}$
12:   $t \leftarrow t + 1$
13: **end while**

---

## B.3 DATASET-WISE UCB1

---

**Algorithm 4** Dataset-wise UCB1

---

1: **Input:** Budget $\mathcal{B}$, input question dataset $\mathcal{D}$ of size $N$, stopping condition $Stop(\mathbf{x}_j)$ for each question $\mathbf{x}_j \in \mathcal{D}$
2: **Input:** $K$ models, the $i$-th model is $f_i$. $\alpha$ is a weight controlling the exploration term.
3: **Environment:** At iteration $t$, a model $\pi_t$ is selected. The cost of using model $f_i$ for question $\mathbf{x}_j$ at time $t$ is $c_{i,t,j}$, and the observed reward is $r_{i,t,j}$. The empirical mean reward of model $f_i$ up to time $t$ is denoted as $\hat{r}_{i,t}$, and the total number of selections of model $f_i$ up to time $t$ is $n_{i,t}$. This applies for all $1 \leq i \leq K$ and $1 \leq j \leq N$.
4: **Initialize:** $t \leftarrow 1$
5: **Initialize:** Remaining budget $\mathcal{B}_t \leftarrow \mathcal{B}$
6: **while** $\mathcal{D} \neq \emptyset$ **do**
7:    **if** $t \leq K$ **then**
8:       $\pi_t \leftarrow t$ {Call each model once in the first $K$ iterations}
9:    **else**
10:      $\pi_t \leftarrow \text{argmax}_i \left( \hat{r}_{i,t} + \frac{1}{\alpha} \sqrt{\frac{2 \ln t}{n_{i,t}}} \right)$ {Select the model that maximizes the upper confidence bound}
11:    **end if**
12:   **for** $\mathbf{x_j} \in \mathcal{D}$ **do**
13:      **if** $Stop(\mathbf{x_j})$ **then**
14:        Remove $\mathbf{x_j}$ from $\mathcal{D}$ {Remove question $\mathbf{x_j}$ that meets the stopping condition}
15:        **continue**
16:      **end if**
17:      Update remaining budget $\mathcal{B}_t \leftarrow \mathcal{B}_t - c_{\pi_t,t,j}$
18:      **if** $\mathcal{B}_t < 0$ **then**
19:        **Exit** {Terminate if budget is exhausted}
20:      **end if**
21:      Use model $f_{\pi_t}$ to generate a response for the question $\mathbf{x}_j$ and observe the reward $r_{\pi_t,t,j}$
22:   **end for**
23:   Update the estimated reward $\hat{r}_{\pi_t,t}$ and the number of pulls $n_{\pi_t,t}$ {Update statistics for the selected model}
24:   $\mathcal{B}_{t+1} \leftarrow \mathcal{B}_t$
25:   $t \leftarrow t + 1$
26: **end while**

---

## B.4 RANDOM MODEL SELECTION

---

**Algorithm 5** Random Model Selection

---

1: **Input:** Budget $\mathcal{B}$, question dataset $\mathcal{D}$, stopping condition $Stop(\mathbf{x}_j)$ for each question $\mathbf{x}_j \in \mathcal{D}$
2: **Input:** $K$ models, where the $i$-th model is $f_i$
3: **Environment:** At iteration $t$, for a given question $\mathbf{x}_j$, model $f_i$ is selected by the action $\pi_t(\mathbf{x}_j)$ (denoted as $\pi_{t,j}$). The observed reward is $r_{i,t} \in [0, 1]$, and the cost is $c_{i,t,j}$.
4: **Initialize:** $t \leftarrow 1$
5: **Initialize:** Remaining budget $\mathcal{B}_t \leftarrow \mathcal{B}$
6: **while** $\mathcal{D} \neq \emptyset$ **do**
7:     **for** $\mathbf{x_j} \in \mathcal{D}$ **do**
8:         **if** $Stop(\mathbf{x_j})$ **then**
9:             Remove $\mathbf{x_j}$ from $\mathcal{D}$ {Remove question $\mathbf{x_i}$ that meets the stopping condition}
10:             **continue**
11:         **end if**
12:         $\pi_{t,j} \leftarrow Discrete\_Uniform(1, K)$ {Select a model randomly from 1 to $K$}
13:         Update remaining budget $\mathcal{B}_t \leftarrow \mathcal{B}_t - c_{\pi_{t,j},t,j}$
14:         **if** $\mathcal{B}_t < 0$ **then**
15:             **Exit** {Terminate if budget is exhausted}
16:         **end if**
17:         Use model $f_{\pi_{t,j}}$ to generate a response
18:     **end for**
19:     $\mathcal{B}_{t+1} \leftarrow \mathcal{B}_t$
20:     $t \leftarrow t + 1$
21: **end while**

---

