# OpenReview forum: "Cost-Effective Synthetic Data Generation for Post-Training using QWICK"
_ICLR.cc/2025/Conference — Submitted to ICLR 2025_

### Official Review · Reviewer_8kVQ · 2024-10-31

**Soundness:** 3
**Presentation:** 2
**Contribution:** 3
**Rating:** 5
**Confidence:** 3

**Summary:**

This paper focuses on addressing the great cost of existing synthetic data generation methods. While stronger models generally produce higher-quality data, they come with significant computational demands. So this paper proposes QWICK, a cost-efficient framework that strategically selects among strong or weak models. By employing a reward mechanism, QWICK identifies the suitable model for each specific question, significantly reducing the cost of generating high-quality data while maintaining strong performance.

**Strengths:**

- The motivation to reduce the cost of generating high-quality synthetic data is compelling. As written in Lines 186–187, models can exhibit varying performance across the entire dataset, so selecting different models is not only appropriate for specific questions but also more cost-effective than relying solely on larger models.
- With the help of budget-limited multi-armed bandits, QWICK considers utility in determining which model to use, the evaluation metrics are the cost of LLM and the reward with ground truth or ORM.
- Experimental results show that QWICK consistently outperforms both the random selection baseline and UCB1 specifically while maintaining sample diversity and coverage.

**Weaknesses:**

- This paper may be challenging for readers unfamiliar with reinforcement learning, including multi-armed bandits. The authors could consider presenting the methods in a more straightforward manner for clarity.
- The authors only test each dataset on models from the same series. Since models from different series could have comparable costs, how would selecting models from different origins impact the results?
- The prompt to generate synthetic data is missing, whether baselines are the same with QWICK.

**Questions:**

please refer to weakness

---

> ### Author Response · Authors · 2024-11-25
>
> We thank the reviewer for the very insightful and helpful comments! We would like to address your questions in the below response.
>
> **Q1: This paper may be challenging for readers unfamiliar with reinforcement learning, including multi-armed bandits. The authors could consider presenting the methods in a more straightforward manner for clarity.**
>
> **A1:** The multi-armed bandit (MAB) problem is a classic framework used to balance exploration and exploitation in decision-making. We have multi-armed bandits and each time we can pull one arm and observe a reward. In the context of our problem, we need to decide which arm to pull to obtain the largest reward over time. In this paper, we formulate each question (prompt) as a bandit and different model as an arm. We need to decide for each question (bandits), which arm can give us the largest reward.
>
> We hope this high-level description adds clarity and we will add it as the opening paragraph of Section 2.2.
>
>
> **Q3: The prompt to generate synthetic data is missing, whether baselines are the same with QWICK.**
>
> **A3:** We maintain a consistent prompt across the baselines and QWICK, with dataset-specific prompts tailored to each task. These prompts are then formatted according to each model's specific chat template. The dataset-specific prompts are as follows:
>
> *Mathematics Dataset:*
> - System prompt: "You are a helpful math solver."
> - User instruction: "Please reason step by step, and provide your final answer within \\boxed{{}}."
>
> *MBPP Dataset:*
> - User instruction combines the programming task with its unit tests, for example:
>
> Write a function to find the sum of first even and odd number of a given list
>
> assert sum_even_odd([1,3,5,7,4,1,6,8])==5
>
> *ToRA Ablation Study:*
> - User instruction includes comprehensive guidelines along with few-shot examples:
>
> Integrate step-by-step reasoning and Python code to solve math problems using the following guidelines:
> Analyze the question and write functions to solve the problem; the function should not take any arguments.
> Present the final result in LaTeX using a \boxed{} without any units.
> Utilize the pi symbol and Rational from Sympy for π and fractions, and simplify all fractions and square roots without converting them to decimal values.
> Here are some examples you may refer to:
>
> Question: Olivia has 23 dollars . She bought five bagels for 3 dollars each. How much money does she have left?
>
> Solution:
> ```python
> def money_left():
>     money_initial = 23
>     bagels = 5
>     bagel_cost = 3
>     money_spent = bagels * bagel_cost
>     remaining_money = money_initial - money_spent
>     return remaining_money
> remaining_money = money_left()
> print(remaining_money)
> ```
> 8
>
> Olivia has 8 dollars left.

---

> > ### Author Response · Authors · 2024-11-26
> >
> > **Q2: The authors only test each dataset on models from the same series. Since models from different series could have comparable costs, how would selecting models from different origins impact the results?**
> >
> > **A2:** Models in different series, even those with comparable computational costs, can exhibit distinct strengths when solving various types of questions, regardless of their model series. Our QWICK framework capitalizes on these specialized capabilities by dynamically selecting the most suitable model for each question, leading to more cost-effective solutions. To validate this approach, we conducted experiments using models of similar size but from different families: Gemma-2 9B, LLaMA-3.1 8B, and Phi-3 7B on the MATH dataset. Our results demonstrate that these models, despite their similar computational requirements, show specialized performance across different question types. In our evaluation, we measured performance using the number of valid solutions generated (i.e., binary reward or diversity), with a cap of 10 generations per problem (same setting as in section 3.1 but with different models). Initially, our method performs slightly lower than single model selection due to necessary exploration costs, but this trade-off enables better long-term performance through optimal model-question matching. The results confirm that leveraging model specialization across different families can obtain much more rewards with the same cost upon a single model and random or dataset-level methods.
> >
> > | Norm. Cost | 1.0 | 2.0 | 3.0 | 4.0 | 5.0 |
> > |----------|-----|------|------|------|------|
> > | **Question Level Bandits (ours)** | 19659 | **39126** | **47849** | **51989** | **54835** |
> > | Random | 18593 | 36489 | 45521 | 49726 | 52734 |
> > | Dataset Level Bandits | 17988 | 36086 | 45191 | 49619 | 52757 |
> > | Gemma2 9B | 18404 | 36748 | 43321 | 46272 | 48296 |
> > | Llama3.1 8B | 17996 | 34089 | 45194 | 50844 | 54344 |
> > | Phi3 7B | **20035** | 38307 | 44062 | 47331 | 49661 |

---

> > > ### Comment · Reviewer_8kVQ · 2024-11-27
> > >
> > > Thank you for the authors' responses, which address some of my concerns.
> > >
> > > However, I am still unclear about the experimental setup regarding Q2. Are the model pools for `Question Level Bandits (ours)` composed of Gemma2 9B, Llama3.1 8B, and Phi 7B? If so, there seems to be a misunderstanding of my comment:
> > >
> > > The original experiments selected models of different sizes from the same series. If the selection is not limited to a single series, why not choose models from the entire pool (different sizes from all series)?
> > >
> > > That said, the additional experiment partially validates this point.

---

> > > > ### Author Response · Authors · 2024-11-28
> > > >
> > > > Dear Reviewer,
> > > >
> > > > Thank you for your feedback. I would like to clarify the experimental setup and results presented in previous response.
> > > >
> > > > In **A2**, the experiments only used a limited model pool consisting of only similar-sized models: Gemma2 9B, Llama3.1 8B, and Phi 7B.
> > > >
> > > > In the following experiments, we expanded the model pool to include a more diverse range of models:
> > > >
> > > > Gemma2: 2B, 9B, and 27B variants
> > > >
> > > > Llama3.1: 8B and 70B variants
> > > >
> > > > Phi3: 7B and 14B variants
> > > >
> > > > Our results demonstrate that even with this expanded and more diverse model pool, the **Question Level Bandits (ours)** continues to outperform all baseline settings in terms of valid solutions generated. This includes both single-model configurations and other model selection methods (Random and Dataset Level Bandits). This finding reinforces the robustness and effectiveness of our proposed method across a broader spectrum of model architectures and sizes.
> > > >
> > > > | Model | 5.0 | 10.0 | 15.0 | 20.0 | 25.0 |
> > > > |-------|-----|------|------|------|------|
> > > > | **Question Level Bandits (ours)** | 17780 | 34307 | **46252** | **51610** | **55467** |
> > > > | Random | 13157 | 24990 | 36043 | 44423 | 49348 |
> > > > | Dataset Level Bandits | 18149 | 25001 | 32690 | 41349 | 47698 |
> > > > | Gemma2 2B | 21108 | 28152 | 31889 | 34505 | 36402 |
> > > > | Gemma2 9B | 18404 | 36748 | 43321 | 46272 | 48296 |
> > > > | Gemma2 27B | 7814 | 15506 | 22622 | 28093 | 35202 |
> > > > | Llama3.1 8B | 17996 | 34089 | 45194 | 50844 | 54344 |
> > > > | Llama3.1 70B | 9726 | 17102 | 23599 | 33139 | 40615 |
> > > > | Phi3 7B | 20035 | 38307 | 44062 | 47331 | 49661 |
> > > > | Phi3 14B| 18802 | 34884 | 41665 | 45376 | 47937 |

---

> > > > > ### Comment · Reviewer_8kVQ · 2024-11-28
> > > > >
> > > > > Thank you for the additional results. This time, the setting aligns with my expectations. However, it appears that directly using Llama3.1 8B, as shown in the table above, achieves performance quite close to those of Question Level Bandits. Could you clarify if there are any other advantages to your approach?

---

> > > > > > ### Author Response · Authors · 2024-11-28
> > > > > >
> > > > > > Dear Reviewer,
> > > > > >
> > > > > > Thank you for your prompt feedback!
> > > > > >
> > > > > > Our proposed method demonstrates superior performance compared to the single Llama-3.1 8B model in two critical aspects:
> > > > > >
> > > > > > Diversity: We achieve a higher number of valid solution generations (as in the earlier response)
> > > > > >
> > > > > > Coverage: As shown in the table below, we solve much more problems from the MATH dataset compared with single Llama-3.1 8B (which contains 7,500 questions in total)
> > > > > >
> > > > > > Coverage (#solved questions)
> > > > > > | Model | 5.0 | 10.0 | 15.0 | 20.0 | 25.0 |
> > > > > > |-------|-----|------|------|------|------|
> > > > > > | **Question Level Bandits (ours)** | 5365 | **6505** | **6607** | **6862** | **6933** |
> > > > > > | Llama-3.1 8B | 5563| 6074 | 6342 | 6518 | 6639 |
> > > > > >
> > > > > > Through this enhanced coverage and diversity, our method provides higher quality synthetic data.
> > > > > >
> > > > > > Furthermore, our method addresses a fundamental challenge: it is very hard to know a priori which model will perform best on a given dataset. Our question-level bandits approach effectively overcomes this limitation by dynamically selecting the most suitable model for each specific problem.

---

### Official Review · Reviewer_N3PA · 2024-11-01

**Soundness:** 3
**Presentation:** 3
**Contribution:** 3
**Rating:** 5
**Confidence:** 3

**Summary:**

This paper presents QWICK (Question-Wise model pICK), a novel algorithm designed to efficiently generate high-quality synthetic data for post-training large language models (LLMs) by dynamically selecting the optimal model for each specific question based on a balance of cost and quality. The approach formulates the problem as a budget-limited multi-armed bandit (MAB), allowing the algorithm to achieve reduction in cost on programming tasks and mathematics datasets compared to traditional methods. QWICK aims to address the challenge of cost-effective synthetic data generation by leveraging both strong and weak models to optimize data quality without exceeding budget constraints.

**Strengths:**

- Cost-Efficiency: The paper proposes a cost-saving solution for synthetic data generation, achieving significant budget reductions without compromising data quality.
- Adaptive Model Selection: QWICK's question-wise approach allows it to adapt to specific data points by selecting models dynamically, which improves data generation's quality and validity across diverse datasets.
- Empirical Validation: The algorithm demonstrates superior performance in experiments compared to baseline methods

**Weaknesses:**

- Dependency on Initial Model Pool: The algorithm's performance is tied to the quality of the initial pool of models, and it does not account for model improvement over time, which may limit its effectiveness in environments with evolving model capabilities.
- Constrained scenario: I assume the authors consider the scenario where the constructer of the dataset uses only API calls. This scenario is kind of narrow, since most of the companies and researchers may have their own GPU resources, I wonder this approach is still applicable under the assumption that some GPU is accessible, so that inference can be done locally.
- What happens to the data generated in previous iterations: I wonder are the samples generated in iterations prior to the termination still kept in the generated dataset? Or are they simply abandoned? If it is the latter, wouldn't it cause waste of computation?

**Questions:**

Please see weaknesses.

---

> ### Author Response · Authors · 2024-11-25
>
> We thank the reviewer for the insightful and helpful feedback! We address all your questions and concerns below.
>
> **Q1: Dependency on Initial Model Pool: The algorithm's performance is tied to the quality of the initial pool of models, and it does not account for model improvement over time, which may limit its effectiveness in environments with evolving model capabilities.**
>
> **A1:** While our current work primarily demonstrates effectiveness with a fixed model pool, our framework is adaptable to scenarios with evolving model capabilities. This is because QWICK maintains a historical performance record for each model-question pair, enabling dynamic updates of model selection as new versions of capabilities emerge. This tracking mechanism allows us to continuously identify the most cost-effective model for each problem type, even as the model pool evolves. Extending our approach to incorporate model fine-tuning based on synthetic data and adapting to evolving model capabilities is an interesting direction for future research.
>
> **Q2: Constrained scenario: I assume the authors consider the scenario where the constructer of the dataset uses only API calls. This scenario is kind of narrow, since most of the companies and researchers may have their own GPU resources, I wonder this approach is still applicable under the assumption that some GPU is accessible, so that inference can be done locally.**
>
> **A2:** Our method extends beyond API-only scenarios. In local deployment scenarios, our algorithm selects the most cost-effective model by optimizing for GPU-hours, where cost is calculated as (GPU cost per second × end-to-end latency), instead of consumed tokens. The rest of the algorithms remains unchangd. This flexibility makes our method applicable to both API-based and local computing environments, helping users optimize resource utilization regardless of deployment strategy.
>
> **Q3: What happens to the data generated in previous iterations: I wonder are the samples generated in iterations prior to the termination still kept in the generated dataset? Or are they simply abandoned? If it is the latter, wouldn't it cause waste of computation?**
>
> **A3:** We implement a quality-based filtering mechanism (as illustrated in the right portion of Fig.1 ) that systematically evaluates all generated data. Data retention decisions are made based on strict quality criteria:
>
> For math tasks: We retain samples where the derived solutions match the correct results
>
> For coding tasks: We preserve examples that successfully pass unit tests.
>
> The primary objective is to collect high-quality reasoning paths that can be used for model fine-tuning.
>
> While we focus on retaining high-quality samples, the “negative” cases (failed attempts) can also be valuable and may be stored separately for specific purposes, such as RLHF. The generation of “negative” cases is inevitable but the proposed QWICK method best allocates the most cost-effective models for each problem and saves budgets.

---

> ### Comment · Reviewer_N3PA · 2024-11-26
> **Response to Authors**
>
> I appreciate the authors' responses during the rebuttal. However, I'm afraid my concerns are not completely addressed at the moment:
> - There is not experiments to support the usage of GPU cost per second × end-to-end latency as the cost during search. In addition, it is unclear how to address the scenario where the user has access to both GPU and API.
> - The value of low quality samples sounds reasonable, however, there is no experiments to support this either.
> Therefore, I believe this paper has proposed a promising technique to create data with low cost, but the authors need to conduct a more thorough investigation on these two points to make the paper more complete. I tend to keep my rating at this moment.

---

> > ### Author Response · Authors · 2024-11-26
> >
> > Dear Reviewer N3PA,
> >
> > Thank you for your thoughtful feedback. We would like to address your remaining concerns:
> >
> > **Q: There is not experiments to support the usage of GPU cost per second × end-to-end latency as the cost during search. In addition, it is unclear how to address the scenario where the user has access to both GPU and API.**
> >
> > **A:** GPU-hours are indeed a well-established metric in LLM training and deployment scenarios. Our approach of multiplying GPU cost per second by end-to-end latency provides a practical approximation of the actual monetary cost for data generation. This metric can be easily converted to a per-token cost basis (i.e., $\text{per token cost} = \frac{\text{GPU price per hour} \times \text{GPU hours}}{\text{Total Generated Tokens}}$), making it directly comparable to API pricing models. For hybrid scenarios where both GPU and API resources are available, our framework naturally accommodates this by converting all costs to a standardized per-token basis, allowing for unified cost optimization across different computation resources.
> >
> > **Q: The value of low quality samples sounds reasonable, however, there is no experiments to support this either. Therefore, I believe this paper has proposed a promising technique to create data with low cost, but the authors need to conduct a more thorough investigation on these two points to make the paper more complete. I tend to keep my rating at this moment.**
> >
> > **A:** While we appreciate the interest in studying low-quality samples, this falls outside the primary scope of our work, which focuses on efficient high-quality data generation. The generation of both high and low-quality samples represents an inherent trade-off in any synthetic data generation process. Our framework's key contribution is minimizing resource waste by selecting the most cost-effective model for each question, which inherently reduces the proportion of low-quality samples compared to baseline approaches. This improvement in efficiency is demonstrated through our experimental results, where we achieve better quality outputs while optimizing resource utilization.
> >
> > Best regards,
> >
> > Paper #2257 Authors

---

### Official Review · Reviewer_MNCQ · 2024-11-04

**Soundness:** 2
**Presentation:** 3
**Contribution:** 3
**Rating:** 5
**Confidence:** 3

**Summary:**

This paper introduces Question-Wise model Pick (QWICK) an approach for generating high quality synthetic data with compute budgets.  QWICK uses a multi-armed bandit strategy to select models that can be used for generation.  The proposed approach is evaluated on mat and coding finetuning benchmarks and results indicate compute improvements from training with QWICK finetuning data.

**Strengths:**

- The paper makes contributions to improving synthetic data generation for finetuning. The authors study this in the setting of limited budget as synthetic data generation is still costly even for finetuning.
- The authors have included algorithms in the paper that detail the implementations including for baseline comparisons in the appendix.  This makes the proposed approach and related approaches easy to understand.

**Weaknesses:**

- There are some missing comparisons including comparison to always using the best and worst models as well as comparisons with real data.  it was unclear to me if this is the red dotted line in the figures as this line was not included in the legend.  Having the results for each model included the figure would also help show how much better the selection is at various amounts of compute versus only using the 2B, 9B, or 27B models respectively.
- It is unclear how strong the models have to be as even the Gemma-2-2B models are already quite powerful, and the effectiveness of the algorithm is contingent on there being a large diversity in the models.  Can the authors include some ablation with a larger variance of models? This also goes in hand with the previous comment on showing the performance of each model individually.
- One limitation of the proposed approach is that the proposed approach is applied to tasks where the answer is easy to verify.  For other tasks (summarization or more complex instruction following), it may be hard to extend the proposed approach as knowing whether a model produces the correct answer is more difficult.
- The experiments only evaluate on train/test with the same data.  It would be interesting to know if the synthetic data has other impact on the evaluation particularly downstream robustness (e.g. train on MATH and test on GSM).

**Questions:**

- What is the red dotted line in the figures?

---

> ### Author Response · Authors · 2024-11-25
>
> We thank the reviewer for the insightful and helpful feedback! We would like to address your questions in the below response.
>
>
> **Q3: One limitation of the proposed approach is that the proposed approach is applied to tasks where the answer is easy to verify. For other tasks (summarization or more complex instruction following), it may be hard to extend the proposed approach as knowing whether a model produces the correct answer is more difficult.**
>
> **A3:** We’d like clairfy that our method extends beyond tasks with deterministic verification, e.g.,  open-ended tasks, as long as there is a (user-defined) reward system to quantify the quality of generated data. In fact, in Math and code, deterministic verification of the results just serves as a simple binary reward system. In more open-ended settings, it can be an outcome reward model set by users.
>
> Note that QWICK always optimizes utility, defined as reward / cost, hence would  remain applicable regardless of the reward type (as explained in section 3.3), whether binary or continuous scores, enabling optimization across diverse task types.
>
> **Q5: What is the red dotted line in the figures?**
>
> **A5:** The red dotted line indicates the speedup of the proposed method mentioned in the text (i.e., speedup to reach the same accuracy).

---

> > ### Author Response · Authors · 2024-11-26
> >
> > **Q1: There are some missing comparisons including comparison to always using the best and worst models as well as comparisons with real data. it was unclear to me if this is the red dotted line in the figures as this line was not included in the legend. Having the results for each model included the figure would also help show how much better the selection is at various amounts of compute versus only using the 2B, 9B, or 27B models respectively.**
> >
> > **A1:** Our analysis below (same setting as in Fig. 4 column 2 but compared with single models) demonstrates that the proposed method significantly improves upon single-model selection by leveraging the specialized capabilities of different models across varied questions. While the Gemma 9B model shows strong overall cost-effectiveness, our experimental results reveal better performance compared with it in both diversity (number of valid generations) and coverage (number of solved questions).
> > For diversity and coverage, our question-level bandit approach initially shows exploration costs, generating fewer valid solutions and solved questions compared to Gemma 9B at a very low cost. However, as the compute budget increases, our method consistently outperforms all baselines, achieving much more diversity and coverage, surpassing both Gemma 9B/2B and random/dataset level selection.
> >
> > Diversity (#valid generations)
> > | Norm. Cost | 1.0 | 2.0 | 3.0 | 4.0 | 5.0 |
> > |----------|-----|------|------|------|------|
> > | **Question Level Bandits (ours)** | 17218 | **37770** | **43621** | **47060** | **49359** |
> > | Random | 12262 | 23261 | 33071 | 39384 | 43554 |
> > | Dataset Level Bandits | 12276 | 17830 | 24997 | 32711 | 39335 |
> > | Gemma 2B | 21108 | 28152 | 31889 | 34505 | 36402 |
> > | Gemma 9B | 18404 | 36748 | 43321 | 46272 | 48296 |
> > | Gemma 27B | 7814 | 15506 | 22622 | 28093 | 35202 |
> >
> > Coverage (#solved questions)
> > | Norm. Cost | 1.0 | 2.0 | 3.0 | 4.0 | 5.0 |
> > |----------|-----|------|------|------|------|
> > | **Question Level Bandits (ours)** | 4828 | 5259 | **5565** | **5748** | **5896** |
> > | Gemma 2B | 3659 | 4148 | 4329 | 4454 | 4513 |
> > | Gemma 9B | 4917 | 5276 | 5507 | 5682 | 5800 |
> > | Gemma 27B | 4562 | 4991 | 5200 | 5388 | 5524 |
> > | Random | 4654 | 5119 | 5407 | 5603 | 5744 |
> > | Dataset Level Bandits | 4619 | 5094 | 5349 | 5464 | 5622 |
> >
> >
> > **Q2: It is unclear how strong the models have to be as even the Gemma-2-2B models are already quite powerful, and the effectiveness of the algorithm is contingent on there being a large diversity in the models. Can the authors include some ablation with a larger variance of models? This also goes in hand with the previous comment on showing the performance of each model individually.**
> >
> > **A2:** We conducted extensive experiments comparing our method across models from different families, with large diversity (i.e., Gemma-2 9B, Llama3.1 8B, and Phi3 7B). While the Llama3.1 8B indeed shows strong performance, our results demonstrate that leveraging model diversity leads to even better outcomes. Despite their similar model sizes, these models exhibit distinct strengths across different question types. Our experimental results show compelling evidence for the effectiveness of our approach with diverse model pools. Our question-level bandit method achieves more valid generations, significantly outperforming both individual models and simpler selection strategies.
> >
> > Diversity (#valid generations)
> > | Norm. Cost | 1.0 | 2.0 | 3.0 | 4.0 | 5.0 |
> > |----------|-----|------|------|------|------|
> > | **Question Level Bandits** | 19659 | **39126** | **47849** | **51989** | **54835** |
> > | Random | 18593 | 36489 | 45521 | 49726 | 52734 |
> > | Dataset Level Bandits | 17988 | 36086 | 45191 | 49619 | 52757 |
> > | Gemma2 9B | 18404 | 36748 | 43321 | 46272 | 48296 |
> > | Llama3.1 8B | 17996 | 34089 | 45194 | 50844 | 54344 |
> > | Phi3 7B | 20035 | 38307 | 44062 | 47331 | 49661 |

---

> > > ### Comment · Reviewer_MNCQ · 2024-12-02
> > >
> > > Thank you for your response to my questions.  Based on the responses, I am maintaining my score.
> > >
> > > - Q1: Based on the comparisons, results are similar for the top performing model are always very close to the results for the Bandit method (< 1% for the Gemma experiments).  It is also unclear to me why the larger model (27B) performs worse than the 9B model on the metrics.
> > >
> > > - Q2: I appreciate the exploration into different model families, but I was wondering more about smaller models (< 2B like 1B and <1B).   Sorry for the ambiguity in the initial question regarding the 2B model already being strong.
> > >
> > > - Q3: I think authors should evaluate the approach on one of these tasks beyond binary (not math/programming) in this case. It is still unclear to me how the results would extend to other tasks (for example translation) particularly as a larger models is much better at this task where the smaller models may be unable to perform well.  This would also be dependent on the model family as well.

---

### Official Review · Reviewer_cE6Y · 2024-11-11

**Soundness:** 2
**Presentation:** 3
**Contribution:** 2
**Rating:** 3
**Confidence:** 5

**Summary:**

This paper addresses the challenge of improving LLMs through high-quality synthetic data generation in a cost-effective way. A common method to boost LLM performance involves sampling reasoning chains from teacher LLMs and fine-tuning the target model on these synthetic samples. However, choosing an optimal teacher LLM is complicated due to two primary factors. First, there is a trade-off between cost and reward: state-of-the-art (SoTA) LLMs produce high-quality reasoning chains but are expensive, while smaller models are cheaper yet prone to errors. Second, different LLMs may excel on specific subsets of data, making model selection question-dependent.

To tackle this, the paper formulates the model selection as a Multi-armed Bandit (MAB) problem with budget constraints, where each LLM functions as an arm of the MAB. It introduces QWICK, an algorithm inspired by the Fractional KUBE algorithm for MABs, which is adapted to choose models on a per-question basis based on their estimated utility (reward/cost) of the choice.

The method is evaluated against several baselines, including random model selection and dataset-level model selection using the popular UCB1 algorithm, which importantly does not consider cost. Experiments on three datasets demonstrate that the proposed approach achieves cost-efficiency and is competitive with baselines while requiring a smaller budget.

**Strengths:**

1. This paper addresses the critical challenge of generating synthetic data to enhance large language models (LLMs) in a cost-effective manner.
2. The proposed algorithm, QWICK, is technically robust, employing a multi-armed bandit formulation with budget constraints.
3. QWICK’s design is simple and has a low memory footprint, facilitating ease of implementation, adoption, and reproducibility.
4. Experimental results show that the method reduces costs by 40-50% compared to baseline methods while maintaining data quality.
5. QWICK demonstrates effectiveness across two distinct reward formulations: binary rewards and ORMs.

**Weaknesses:**

The key weaknesses of the paper can be grouped into the following broad themes:

---
### Methodological Limitations
---

1. **Sparse Data and Inefficiency**: QWICK tracks raw rewards and costs for each (question, LLM) pair individually, using average reward and cost estimates per pair for model selection. This approach has limitations, as empirical data required for reward and cost estimation can become sparse, particularly as the number of questions grows, the LLM pool expands, or the budget decreases. Furthermore, the algorithm is inefficient, as it estimates reward and cost for each question in isolation without leveraging similarities across questions. A more effective approach might involve using dense representations of questions and training reward and cost models for each LLM.

&nbsp;

---
### Inadequate Experimental Validation
---

The paper's experimental design lacks critical baseline comparisons, raising questions about the benefits of the proposed approach:

2. **Lack of Individual Model Baselines**: Current comparisons focus only on model selection approaches. Benchmarking against baselines that generate synthetic data by sampling from individual models in the pool (e.g., the weakest, strongest, etc.) within budget constraints would clarify whether model selection truly adds value or if a single model could achieve comparable performance.

3. **Question-Wise UCB1 Comparison**: The paper only benchmarks against a dataset-level UCB1 approach, where UCB1 selects a single model for the entire dataset. However, UCB1 could also be adapted for query-level model selection by using the cost term from Algorithm 1 solely to determine when to stop generating rather than to select between models. A comparison of QWICK with this question-level UCB1 algorithm would provide a fairer assessment of the benefits of incorporating cost into the model selection process.

4. **Baselines Incorporating Model Performance Priors**: LLMs often come with prior information about their general or task-specific performance, which can inform model selection by using this information as a reward prior. To evaluate QWICK more thoroughly, it should be compared against simple baselines that leverage this performance data. For instance, rather than sampling models uniformly at random (as done in the Random baseline), models could be sampled from a weighted distribution that reflects their relative performance, whether general or task-specific.

5. **Missing details**: Key experimental details are missing in the paper, such as:
    - *Diversity and Coverage Metrics*: Although these metrics are referenced throughout the paper, they are not formally defined, leaving their specific calculation and interpretation unclear.
    - *Budget Interpretation*: The paper employs normalized budgets to compare methods but lacks a clear, interpretable context for these values. For instance, it is unclear whether a normalized budget of 1 represents the resources needed for a single pass over the entire dataset by the weakest LLM, the strongest LLM, all LLMs combined, or multiple passes with these models. Without this grounding, it’s challenging to gauge the flexibility or limitations of a given budget setting, and to determine which additional baselines would be relevant for comparison.

&nbsp;

---
### Unclear Broader Impact
---
The broader impact and applicability of the paper’s approach are unclear, as it is evaluated within a narrow context: generating Chain-of-Thought (CoT) responses from a limited pool of LLMs for fine-tuning. This limited scope raises questions about the approach's generalizability and whether QWICK is the optimal method for synthetic data generation under resource constraints.

6. Specifically, the paper lacks comparison to alternative methodologies that could also offer an efficient use of the data generation budget, such as:
- *Advanced Inference Techniques*: More sophisticated (and costlier) methods, like Monte Carlo Tree Search (MCTS) or self-refinement, could be used to generate higher-quality responses rather than relying on standard CoT sampling.
- *Focusing on Challenging Tasks*: Instead of generating numerous correct answers for simpler tasks, prioritizing accurate solutions for more challenging tasks could make better use of resources, especially when combined with advanced inference techniques.
- *Training with Other Preference Learning Approaches*: Preference learning methods like Direct Preference Optimization (DPO) can utilize incorrect samples, which QWICK currently discards. Comparing QWICK’s performance when training with DPO rather than Supervised Fine-tuning (SFT) on its generated data, alongside appropriate DPO baselines, would provide a more thorough evaluation of its cost-effectiveness.

Comparison with these alternative approaches would offer a clearer view of QWICK’s relative strengths and limitations, contributing more substantially to understanding best cost-effective synthetic data generation practices for fine-tuning LLMs.

&nbsp;

---
### Ablation and Scaling Experiments
---
The paper lacks ablation experiments on various components of the algorithm, and its scalability remains uncertain.

7. **Question-Level Rewards Ablation**: It would be insightful to ablate the question-level rewards in QWICK, which can get sparse, and rely solely on dataset-level rewards for model selection. This comparison would clarify the necessity of question-level reward tracking.
8. **Dataset-Level Rewards Ablation**: Similarly, removing dataset-level rewards and evaluating its impact on overall performance would provide useful insights.
9. **Scaling the Number of Models**: The current experiments limit QWICK to selecting among a maximum of three models from the same family. Testing QWICK’s scalability with a larger and more diverse model pool could reveal how it handles sparser rewards and increased costs, and may also help identify an optimal range for the number of candidate models.
10. **Scaling the Dataset Size**: Evaluating QWICK’s performance on larger datasets could yield valuable insights as well, as this would also lead to sparser rewards and increased costs.

**Questions:**

Please see the *Weakness* section for recommendations on experiments and analyses that could further enhance the paper.

**Additional questions and suggestions (not affecting the score)**:
1. How are the initial values for rewards, costs, and usage counts set for the first model in the model pool in Algorithm 1?
2. The question-level reward term occasionally lacks the question number subscript `j` (e.g., on line 136).
3. How is the cost term determined in line 25 of Algorithm 1? Should this line perhaps follow line 29? A similar issue appears in other algorithms within the appendix.
4. Why is the dataset-level reward term updated after each question in Algorithm 1 (line 30), rather than after completing the entire dataset? Has this design choice been empirically validated?
5. Is there an explanation for the observed decrease in model performance when the budget is increased for the MATH and MBPP datasets in Figure 4?

---

### Meta-Review · Area_Chair_2Lah · 2024-12-23

**Metareview:**

This paper studies a very valid problem: balancing the quality and cost tradeoff of synthetic data generation used for post-training. The author introduces Question-Wise model Pick (QWICK), an approach for generating high quality synthetic data with compute budgets.
QWICK uses a budget-limited question-wise multi-armed bandit (MAB) framework to balance exploiting well performing models and exploring less-utilized ones. The proposed approach is evaluated on math and coding finetuning benchmarks, on a variety of open-sourced models, and results indicate compute improvements from training with QWICK finetuning data.

While the paper explores an important question (reducing cost while maintaining quality for data generation), and demonstrated promising results, it still poses challenges for implementation, such as requiring a defined reward for each question. The evaluation can also be improved by having comparisons to, say, real data.

**Additional Comments On Reviewer Discussion:**

It was a generally healthy discussion. One reviewer (the lowest rating one) appears to be using LLM generated reviews, so the author chose not to engage. I discounted that particular review when making the final recommendation.

---

### Decision · Program_Chairs · 2025-01-22

Reject